# Optochemical control over mRNA translation by photocaged phosphorodiamidate morpholino oligonucleotides in vivo

Katsiaryna Tarbashevich[1], Atanu Ghosh[2], Arnab Das[2], Debajyoti Kuilya[2], Swrajit Nath Sharma[2], Surajit Sinha [2] ✉ & Erez Raz [1,3] ✉

We developed an efficient, robust, and broadly applicable system for light-induced protein translation to control the production of proteins of interest and study their function. The method is based on the displacement of a single type of antisense morpholino from RNA by the uncaged guanidinium-linked morpholino (GMO)-phosphorodiamidate morpholino oligonucleotide (PMO) chimera upon UV irradiation. The GMO-PMO chimera designed here is cell-permeable and the GMO part can be produced employing a mercury-free approach compatible with the synthesis on solid support. We demonstrate the function of this optochemical approach in live embryos by inducing, at desired times and locations, the expression of proteins that label specific cells, ablate tissue regions, and affect embryonic development. Together, our results demonstrate that the cell-permeable GMO-PMO chimera offers a strategy for controlling the function of mRNAs of interest. This method allows for the production of proteins at specific times and positions within live organisms, facilitating numerous applications in biomedical research and therapy.

The ability to control protein expression in a spatiotemporal manner in vivo is a powerful tool for research in the fields of developmental and cell biology, as it can be used to regulate embryo development, organoid patterning, and stem cell differentiation (reviewed in refs. 1,2). The existing light-based promoter induction methods necessitate the generation of transgenic embryos for each gene of interest and suffer from a significant time delay between promoter induction and protein translation[2–5]. Consequently, for studying rapid biological processes, RNA-based methods are preferable. Current methods for inducing protein translation of exogenously introduced RNA at specific times and locations rely on introducing into cells modified RNA that contains synthetic 5'Cap analogs with photo-responsive groups[6,7]. Upon irradiation, the translation machinery

interacts with the RNA, and the translation of marker proteins such as GFP and Luciferase can be induced in cells in culture and in zebrafish embryos[6–8]. A biological effect was obtained when employing this approach under in vitro conditions, where induced H-Ras expression resulted in neurite expansion[6]. Thus far, no reports have shown that this system can function in affecting biological pathways and developmental processes in whole animals. An additional strategy for controlling RNA translation employs morpholino oligonucleotide (MO) which binds RNA targets and inhibits the binding of ribosomes, a technology that constitutes a widely used tool for analysis of protein function in vivo[9]. Controlling the activity of MO optochemically was first presented by Shestopalov et al. 2007[10], after which researchers offered a range of designs for inhibitory photocaged structures[11–13].

[1]Institute of Cell Biology, Center for Molecular Biology of Inflammation (ZMBE), Münster, Germany. [2]School of Applied and Interdisciplinary Sciences, Indian Association for the Cultivation of Science, Jadavpur, Kolkata, India. [3]Max Planck Institute for Molecular Biomedicine, Münster, Germany. ✉ e-mail: ocss5@iacs.res.in; erez.raz@uni-muenster.de

While this technology is effective, it requires an individual design for each target and, thus far, is not commonly used. Critically, these approaches are geared toward inhibiting translation rather than the induction of translation. Further, although a MO-based method was developed for turning on RNA translation by light-induced inactivation of a MO, allowing translation to start[13], synthesizing such photo morpholinos can be challenging, as it requires special photolinker subunits for chain elongation and calibration of MO lengths. As such, following the original demonstration, the methodology is not employed for turning on RNA translation.

To address these shortcomings, we and others have previously established automated platforms for synthesizing phosphorodiamidate morpholino oligonucleotides (PMOs)[14–16] and introduced several modifications in PMOs to increase duplex stability and also hydrophobicity[17]. We have also incorporated four guanidinium linkages in PMOs (resulting in GMO-PMO chimeras) to enhance this molecule's cellular uptake compared to unmodified PMO[18]. In the current work, we combined these two approaches to establish a robust method for light-induced initiation of mRNA translation by photocaged GMO-PMOs and developed a method that eliminates the need for mercury in the synthesis of GMO. Importantly, the method is applicable for the synthesis of guanidinium linkage with the secondary amine, which was otherwise challenging to perform under mild conditions. We demonstrate the power of this technology, which is based on the controlled displacement of a translation-blocking MO antisense oligonucleotide from mRNA, to turn on translation in live embryos at specific times and locations of interest. The procedure we present allows for labeling structures of interest with fluorescent proteins, eliminating specific cells, and influencing embryonic development by controlling the expression of signaling molecules, thereby contributing to the chemical toolkit available for basic and biomedical research using zebrafish and other in vivo models[19].

## Results

The protein expression strategy we established relies on strand displacement of the target mRNA from a complex with the translation-blocking morpholino (tbMO). The displacement of the RNA occurs by an uncaged photomorpholino (PMO). The system thus comprises a standard Morpholino (Gene Tools, hereafter referred to as tbMO, indicated by the black bar at the bottom left cartoon in Fig. 1) that targets a specific RNA sequence (represented by the yellow bar in Fig. 1), and a photocaged PMO (cPMO, depicted by the yellow striped bar in Fig. 1). Uncaging the PMO that has a sequence complementary to the tbMO leads to the binding of the two and the release of the RNA for translation. Importantly, the control morpholino and the reagents for synthesizing the RNA in vitro are commercially available. Further, the photocaged PMO used in the different experiments presented below has a specific non-variable sequence. All the morpholinos can be synthesized using a protocol standardized in an automated oligo synthesizer[14].

For the GMO-PMO chimeras, the GMO part was initially synthesized manually on solid supports employing a mercury-free approach. For that, we optimized the GMO synthesis in solution. First, we attempted to couple 7′-Tertbutyldiphenyloxy morpholino uridine (TBDPS-NH) with 7′-fluorenylmethyloxycarbonylthiourea morpholino thymidine monomer (FmocNCS) (1 and 2, respectively in Fig. 2) using $I_2$ and 2,2,6,6-tetramethylpiperidine (TMP) in (3:1) acetonitrile (ACN) / dichloromethane (DCM), following the procedure of deoxynucleic guanidine (DNG) synthesis reported by Skakuj et al.[20]. However, the FmocNCS thymidine monomer was insoluble in this solvent system, prompting us to perform the reaction in DCM solvent. In addition, we found that the TMP was forming adducts as the major side products with the carbodiimide intermediate, reducing the efficiency of TBDPS-NH uridine reaction (mass analysis Supplementary Fig. 1). Unlike previously published DNG synthesis[20], in this case, secondary amine was

involved. However, the synthesis of guanidinium linkage is a long-standing challenge if secondary amine is used as the coupling partner[21]. This issue is even more significant if Fmoc- or carbamate-protected thiourea is used instead of amide protection, where the electrophilic character of thiourea can be increased.

Furthermore, GMO-PMO chimera synthesis was carried out on ACN incompatible polystyrene resins instead of controlled pore glass (CPG) support used for DNG synthesis[20]. To the best of our knowledge, methods compatible with the synthesis of GMO have thus far not been developed[22]. To address this issue, we explored non-nucleophilic bases that are compatible with the Fmoc group, including pyridine (Py), triethylamine (TEA), 1-methylimidazole (NMI), N-ethylmorpholine (NEM), 1,4-diazabicyclo [2.2.2] octane (DABCO) in DCM as a solvent (Fig. 2) and at the same time Fmoc can be deprotected under mild conditions keeping GMO-PMO intact. Moreover, polystyrene has a good swelling property in DCM. Among these, NEM and DABCO provided the best results. However, DABCO and iodine formed a dark brown precipitate (Supplementary Fig. 2), prompting us to proceed with NEM. We observed that the large excess of NEM increased the product yield without affecting the Fmoc group (Table in Fig. 2). To further validate this solution phase screening, we then synthesized 5-mer-T GMO on Ramage ChemMatrix resins manually (Supplementary Fig. 3), which was then transferred to the oligo synthesizer for the synthesis of the PMO part to obtain a 10-mer GMO-PMO chimera (GMO-PMO-1), that consists of 4 guanidinum and 5 phosphorodiamidate linkages (Supplementary Fig. 4). In addition, we validated the method by synthesizing GMO-PMO of A and T nucleobase mix sequence (GMO-PMO-2). Following these protocols, several photocaged GMO-PMO and PMO (as control) sequences required for this study were synthesized (Supplementary Figs. 3, 4 and Supplementary Table 1). The GMO part was synthesized manually in order to avoid the blockage of synthesizer tubes due to the Sulfur precipitation. Our GMO-PMO chimera synthesis by semi-automated oligosynthesizer is an approach that was not previously reported on. Incorporation of 5-mer GMO into PMO was necessary to ensure the cell-permeability of GMO-PMO chimera, as we have previously reported on[18]. All oligos were purified by HPLC and characterized by MALDI-TOF (Supplementary methods and Supplementary Tables 3 and 4). For the in vitro and in vivo localization studies, BODIPY-conjugated oligos were synthesized (Supplementary methods, Supplementary Fig. 5).

In contrast to the regular PMO (cPMO1, Supplementary Table 1), we based the synthesis of our photocaged PMO on the GMO-PMO chimera design (cPMO2), which has the same sequence as that of PMO but exhibits better cell permeability[18]. To further increase lipophilicity, we incorporated the phenylacetylene-modified C (C, Table 1, green in Fig. 1 and Supplementary Table 1). We then referred to this phenylacetylene-GMO-PMO as cPMO2. As presented in Figs. 3–7, the translation-blocking morpholino (tbMO) inhibits RNA translation. Upon UV irradiation, the cPMO2 is activated and binds the translation-blocking morpholino, which, in turn, is displaced from mRNA, thereby allowing mRNA translation.

Before testing the system in the developing embryo, we examined the duplex stability by measuring the thermal melting temperature ($T_m$; Table 1, Supplementary Figs. 6, 7) of the control translation-blocking morpholino (tbMO) with RNA, regular PMO1 with tbMO, GMO-PMO (PMO3) with tbMO and the cPMO2 with tbMO before and after 365 nm UV exposure (Supplementary Figs. 6, 7). In the presence of three photocaging groups in cPMO2 (T, Table 1, pink in Fig. 1), the duplex was destabilized by 10 °C, as compared to the non-caged PMO2. The $T_m$ values were comparable for the duplex of tbMO with both PMO1 and PMO3/PMO2 (Table 1).

We also analyzed the global conformation of the duplexes by CD-spectroscopy (Supplementary Fig. 8) and observed that they exhibit characteristic bands. The tbMO-RNA duplex showed a B-type helical structure with a maximum of 265 nm and minima of 245 and 210 nm. A

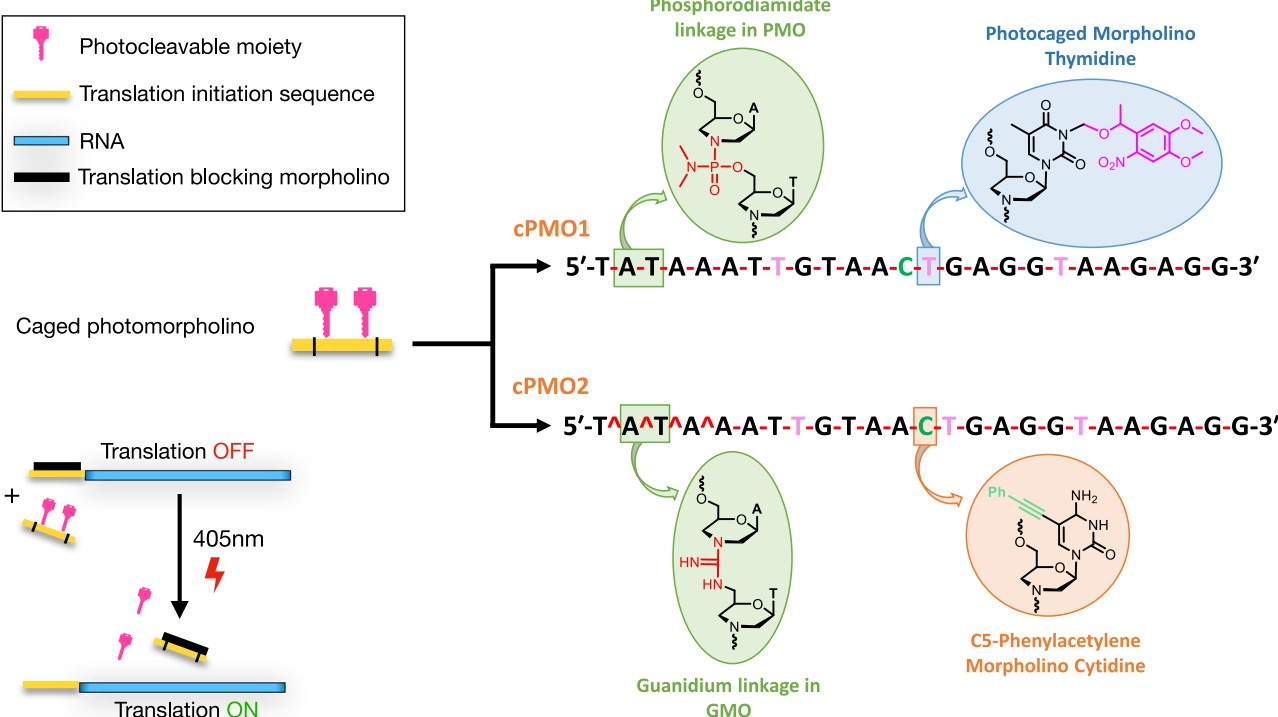

**Fig. 1 | Photomorpholino used for light-induced protein expression.** cPMO is a photocaged morpholino oligonucleotide where three photocaging groups are attached (pink color), and the morpholino unit is linked by a phosphorodiamidate linkage (pink lines, -), and also a regular cytidine is replaced by C5 phenylacetylene–modified cytidine. In cPMO2, the first four linkages are replaced by guanidinum (pink circumflex, ^), and the rest of the linkages constitute the phosphorodiamidate backbone. Translation-blocking morpholino sequence: 5'-CCTCTTACCTCAGTTACAATTTATA-3' (tbMO).

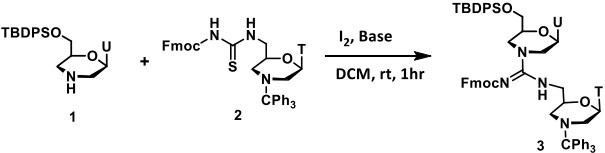

| Entry | Base | Equiv. of base | Product yield (%) |
|-------|------|----------------|-------------------|
| 1 | TMP | 3.5 | 40 |
| 2 | Pyridine | 3.5 | 22 |
| 3 | TEA | 3.5 | 50 |
| 4 | NMI | 3.5 | 60 |
| 5 | DABCO | 3.5 | 62 |
| 6 | NEM | 3.5 | 70 |
| 7 | NEM | 10 | 83 |

**Fig. 2 | Optimization of mercury-free dimer-GMO synthesis.** Compound 2 (7'-Tertbutyldiphenyloxy morpholino uridine) was dissolved in dry DCM followed by sequential addition to the reaction of iodine, base and of Compound 1 (7'-fluorenylmethyloxycarbonylthiourea morpholino thymidine). After completion of the reaction, solvent was removed and the organic layer was extracted from ethyl acetate, concentrated in vacuo and purified. For detailed procedure see Supplementary Methods (Synthesis of the GMO dimer). TMP 2,2,6,6-tetra-methylpiperidine, TEA Et3N, NMI 1-methylimidazole, NEM N-ethylmorpholine. The primary hydroxyl group of morpholino is mentioned 7' for easy understanding.

showed a B-type helix and redshift with a maximum of 274 nm. In summary, the overall global structures of all duplexes are similar, which facilitates displacement of antisense tbMO by PMO2.

To establish a robust, broadly applicable and reproducible method employing the developed oligonucleotides, we designed a plasmid (termed pTIS for plasmid with Translation Initiation Sequence, Fig. 3a) that includes a promoter for in vitro transcription (green in Fig. 3a) followed by a binding site for a specific translation-blocking morpholino oligonucleotide used throughout this work (yellow, here-after called the "translation initiation sequence") and several restriction enzyme cloning sites for DNAs encoding for any protein of interest (blue in Fig. 3a). In vitro transcription can then be employed to produce mRNAs whose function can be spatiotemporally controlled (Fig. 3b). In the experiments themselves we injected zebrafish embryos with mRNA encoding for different proteins, together with two oligonucleotides, namely a standard translation-blocking morpholino (tbMO, standard control Morpholino (Gene Tools), black) and the morpholino con-taining the light-sensitive moieties (cPMO2, yellow with black bars and pink moieties). The translation-blocking morpholino is complementary to the constant translation initiation sequence in the mRNA (yellow) and, therefore, initially blocks the translation of the transcript. The cPMO2 has the same sequence as that of the translation initiation sequence and, thus, is complementary to the translation-blocking morpholino. However, the photocleavable moieties first prevent the translation-blocking morpholino from binding to the cPMO2 (Fig. 3b, upper arrow). Upon UV irradiation (Fig. 3b, lower part), the PMO2 is "activated" (i.e., the photosensitive moieties are cleaved off), such that PMO2 can then sequester the translation-blocking morpholino and relieve the inhibition over translation.

## Induced expression of a fluorescent protein

To test our method in the context of a live developing embryo, we first examined its efficiency in inducing the local translation of a fluorescent

similar helix was observed for the tbMO-PMO1 duplex with maxima at 261 and 225 nm (Supplementary Fig. 8). However, GMO-PMO (PMO2 and PMO3) with tbMO duplexes showed a positive Cotton band compared to the tbMO-RNA duplex. The tbMO-cPMO2 duplex also

**Table 1 | Thermal melting temperature of duplexes. Structures and linkages of the oligos are shown in Fig. 1**

| Oligo | Sequence | With complementary tbMO $T_m$ (°C) |
|---|---|---|
| RNA | 5'-uauaaauuguaaaugagguaagagg-3' (target RNA sequence) | 48 |
| PMO1 | 5'-TATAAATTGTAA**C**TGAGGTAAGAGG-3' | 36 |
| PMO2 | 5'-T^A^T^A^AATTGTAA**C**TGAGGTAAGAGG-3' | 34 |
| PMO3 | 5'-T^A^T^A^AATTGTAACTGAGGTAAGAGG-3' | 37 |
| cPMO2 | 5'-T^A^T^A^AA**T**TGTAA**C**TGAGG**T**AAGAGG-3'(before UV exposure) | 24 |
| PMO2 | 5'- T^A^T^A^AATTGTAA**C**TGAGGTAAGAGG-3' (after UV exposure) | 35 |

Measurements were performed in 40 mM phosphate buffer (pH 7) with a concentration of 2 μM PMO (each strand) with complementary tbMO sequence (5'-CCTCTTACCTCAGTTACAATTTATA-3'). $T_m$ values reported are the averages of two independent experiments that were within ±1.0 °C. Tm values were calculated from the first derivative plot.

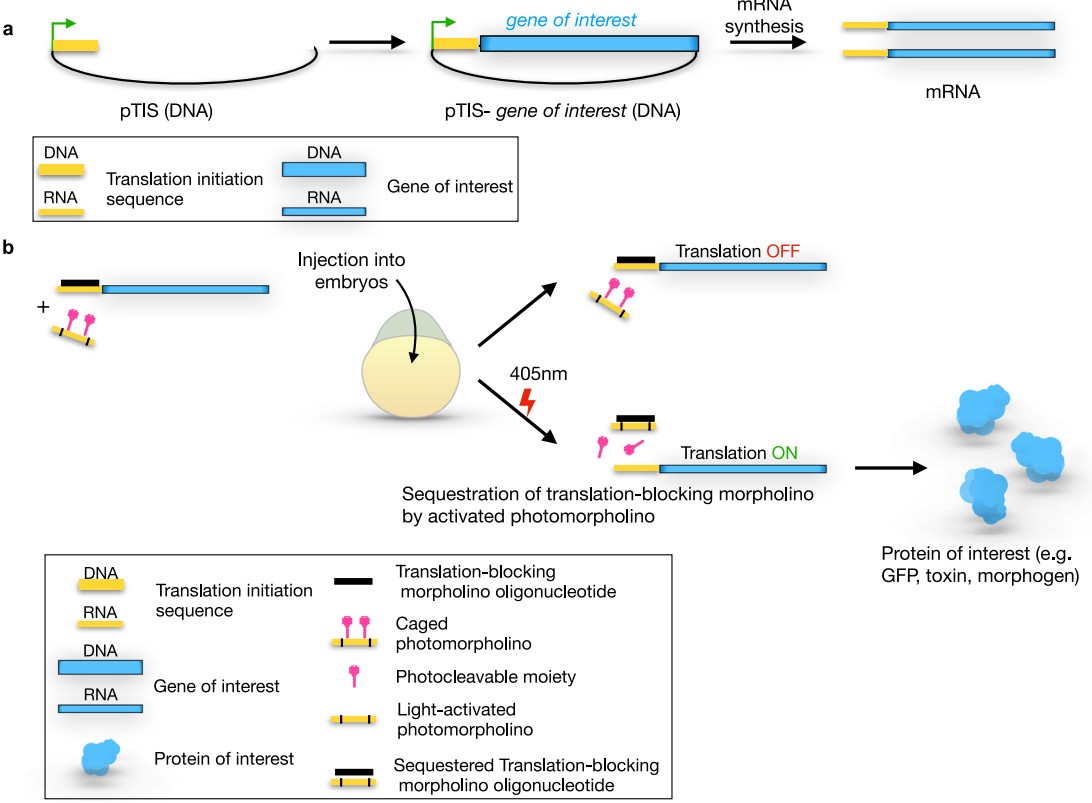

**Fig. 3 | The structure of the pTIS plasmid and general experimental scheme of light-induced protein expression in zebrafish embryos. a** The pTIS contains the promotor for the in vitro transcription (green arrow) and the translation initiation sequence (TIS, yellow), which constitutes the binding site for the standard translation-blocking morpholino. Following subcloning the DNA sequence encoding for the protein of interest (blue), the mRNA is synthesized in vitro. The transcripts obtained in this procedure are used for the light-inducible protein expression. **b** Zebrafish embryos of the 1-cell stage are injected with reporter mRNA (yellow-blue) and two oligonucleotides – the standard translation-blocking morpholino (black) and the photomorpholino (yellow-pink). Reporter mRNA contains TIS (yellow) such that translation is inhibited upon binding of the translation-blocking morpholino (black). The photo morpholino bears the same sequence (yellow), but its binding to the translation-blocking morpholino is prevented by the presence of light-sensitive moieties (pink). Before laser irradiation, the translation of the reporter is blocked by the translation-blocking morpholino (up-pointing arrow), upon light irradiation (405 nm, lower arrow), the light-sensitive moieties (pink) are cleaved, and the translation-blocking morpholino is sequestered from the reporter mRNA by binding to the activated photomorpholino enabling translation from the reporter transcript (blue, lower panels).

protein (Fig. 4). To this end, we cloned the open reading frame of a nucleus-targeted green fluorescent protein (GFP) into the pTIS plasmid. Zebrafish embryos were injected with mRNA produced from this construct together with the translation-blocking morpholino (black) and the "inactive" cPMO2 (yellow + pink) (Fig. 4). In such embryos, translation of the nuclear GFP transcript was inhibited by the blocking morpholino, and no GFP signal could be observed (Fig. 4a and Supplementary Fig. 9). Conversely, upon local irradiation PMO2 was activated, such that it sequestered the translation-blocking morpholino, leading to the translation of GFP and its accumulation within the nuclei (green in Fig. 4b and Supplementary Fig. 9).

Employing this experimental scheme, we found that displacement of the translation-blocking morpholino from the RNA, as judged by the induction of translation upon irradiation, took place when cPMO2 was used, but the displacement was not as efficient under the same experimental setup with regular cPMO1 (Only 45% of irradiated embryos injected with cPMO1 exhibited local GFP expression, in comparison to 89% of those injected with cPMO2). This was also evident from the gel electrophoresis of the duplexes in the presence of PMOs: The presence of PMO1 resulted in a gradual decrease of the duplex intensity in a dose and time-dependent manner, and PMO2 (GMO-PMO) was more effective. Specifically, at the same PMO1 and

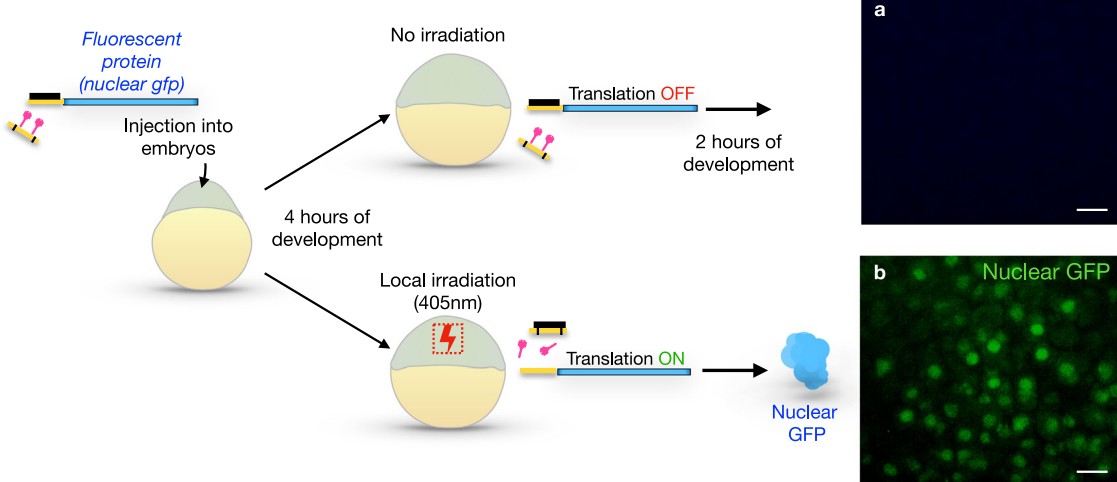

**Fig. 4 | Light-controlled expression of a fluorescent protein in zebrafish embryos.** Zebrafish embryos were injected with the mRNA encoding for the nuclear GFP (blue), cPMO2 (yellow with black stripes and light-cleavable moieties in pink), as well as with the standard translation-blocking morpholino (black). Embryos were allowed to develop until 4 hours post fertilization (hpf). **a** Some of the embryos were kept in the dark (upper arrow), such that the translation of the mRNA was blocked by the translation-blocking morpholino. In this case, no expression of nuclear GFP protein was detected (upper right panel). **b** Sibling embryos were subjected to local irradiation with the 405 nm laser (lower panels, red dotted box), resulting in the photolysis of PMO's light-sensitive moieties. This procedure resulted in sequestration of the translation-blocking morpholino, thus allowing the translation of the nuclear GFP, which was visualized 2 h post-irradiation (green spots). Scale bar 20 µm. Source data are provided as a Source Data file.

PMO2 concentrations, PMO1 produced a duplex intensity of about 43%, while being more effective, PMO2 yielded about 30% duplex intensity relative to the positive control (Supplementary Fig. 10).

In addition to the increased potency in displacing the tbMO, the synthesized GMO-PMO should exhibit improved diffusion throughout the embryo as compared with the standard PMO1[11]. To examine this possibility directly, we performed a flow cytometry experiment using BODIPY-tagged PMOs (Supplementary Fig. 11) in the HeLa cell line, examining the effect of the oligonucleotide dose and the inculcation time. This was further verified by fluorescent microscopy imaging in live cells. We observed PMO2 had superior cell permeability as compared with PMO1, with the signal increasing with time. Next, we examined this point under in vivo conditions in live embryos. Here, we introduced BODIPY488-tagged PMO2 (Supplementary Figs. 4, 5) and PMO1-BODIPY488 oligonucleotides (Supplementary Figs. 4, 5) as in the previous experiment (Supplementary Fig. 11) and another GMO-PMO (PMO3) lacking the Phenylacetylene residue (Supplementary Fig. 4) together with cell-impermeable Dextran-680 into a 1-cell-stage embryo that was allowed to develop until the 1000-cell stage. Subsequently, cells from these embryos were transplanted into host embryos expressing membrane-targeted mCherry protein (Supplementary Fig. 12a, b–b″). Consistent with this supposition, we observed significantly better transfer of the tagged GMO-PMOs (PMO2/3-BODIPY). This was most obvious for PMO2 that was synthesized employing the procedure we developed in this work as determined by the transfer out of Dextran-680–containing cells into cells in their vicinity (Supplementary Fig. 12c). Based on these results, in the following experiments, we employed cPMO2.

## Local induction of the toxin expression

Being able to locally eliminate certain cell groups is an important tool, especially for studies in developmental biology and regeneration. To examine whether our system can induce cell death at a specific time and location within developing tissue, we aimed at controlling the expression of a potent toxin (Kid[23,24]) by light. Here, we followed a procedure similar to that used for GFP expression: we cloned the open reading frame encoding the Kid protein into the universal pTIS plasmid and injected the RNA produced from this vector into embryos together with different morpholino oligonucleotides (Fig. 5a–c). Inhibition of the translation of the Kid-encoding RNA by the tbMO allowed normal development of the embryos that were subjected to UV irradiation, meaning that irradiation by itself does not affect cell survival (Fig. 5a–a″). Including the cPMO2 without irradiating the embryos led to the same results (Fig. 5b–b″). In contrast, including PMO2 in the mix and activating it by light relieved the inhibition of translation, resulting in toxin expression and cell death within the irradiated region (Fig. 5c–c″). The number of apoptotic cells was monitored by detecting the activated caspase-3 protein, a marker of apoptosis[25] (Fig. 6).

## Local induction of the morphogen expression

Proper development of tissues, organs, and whole animals involves interactions and communication among cells, often mediated by secreted signaling molecules. Accordingly, gradients of proteins that provide cells with positional information are key for the establishment of embryonic polarity (e.g., Wnt/β-catenin and BMP[26,27], reviewed in ref. [28]). To examine the potential of our system for directing cell differentiation in an in vivo context, we induced the expression of either bone morphogenic protein (Bmp2b)[29] or β-catenin (Ctnnb1)[30] by light (Fig. 7 and Supplementary Figs. 13, 14).

Indeed, induction of Bmp2b expression by light impaired the development of dorsal tissues, with a pronounced loss of head structures, as expected from an activity that antagonizes the formation of dorsoanterior tissues (Fig. 7a–a″, Supplementary Fig. 13b–e). Importantly, control embryos that were not injected with cPMO2 and irradiated developed normally (Supplementary Fig. 13a–a″), showing that the irradiation per se did not affect embryo development.

Conversely, as manifested by the strong increase in expression of a dorsal mesoderm marker gene (no tail[31],), induction of β-catenin by light-directed embryonic cells to assume this identity[30] (Fig. 7b, b″–d). Therefore, the caging moieties that are released upon light irradiation do not affect the ability of cells to transcribe RNAs of endogenous genes. The specificity of the treatment was additionally controlled by performing experiments without irradiation or without including the cPMO2. In these control cases, the embryos developed normally (Fig. 7b′ and Supplementary Fig. 14).

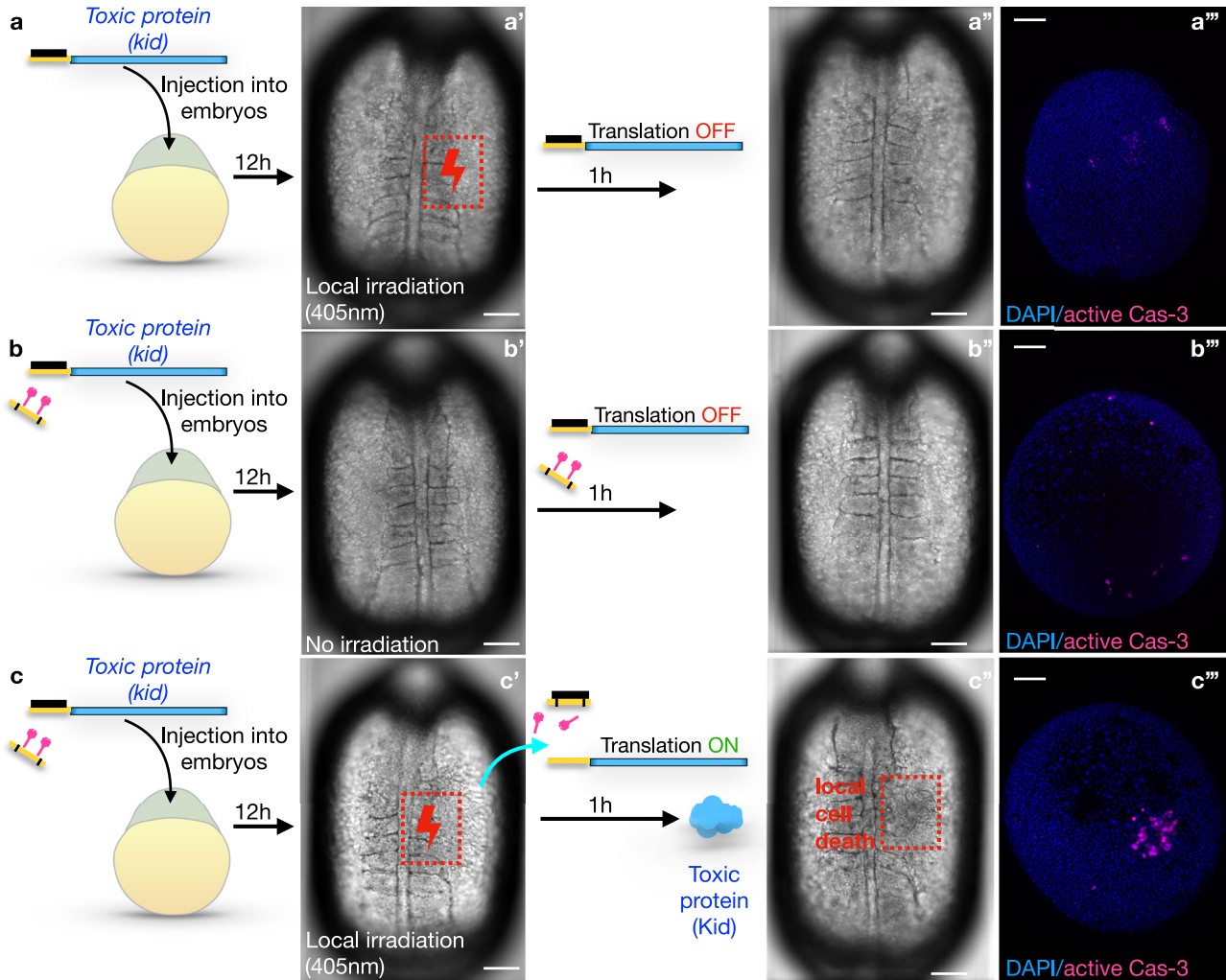

**Fig. 5 | Light-controlled elimination of a specific embryonic cell population by expression of a toxic protein.** Zebrafish embryos were injected with mRNA encoding for the Kid toxin (blue), and with the standard translation-blocking morpholino (black) only (**a**) or with the same mix together with the cPMO2 (yellow with black stripes and pink shapes) (**b**, **c**). Embryos were then allowed to develop for 12 h. At this stage, some embryos (**b - b'''**) were kept in the dark, such that the translation of the mRNA was blocked by the translation-blocking morpholino, and no toxin was produced. Other embryos (**a– a'''**, **c–c'''**) were subjected to 405 nm laser irradiation at a specific region of the embryo (red dotted boxes). Laser irradiation in the absence of the PMO2 had no effect (**a–a'''**). Irradiation in the presence of the PMO2 (**c–c'''**) resulted in the photolysis of PMO2's light-sensitive moieties, sequestration of the translation-blocking morpholino, translation of the RNA, and production of the toxin. Cell apoptosis was detected only in the irradiated region by 1 hour post-irradiation (**c''** red box, and **c'''**). (**a'''**, **b'''**, **c'''**) Embryos were subjected to immunostaining for the active caspase-3, a marker of apoptosis. The embryos in these panels are not the same as those presented in the bright-field panels left of them. Scale bar 150 μm.

Thus far, our system can be used for testing the immediate responses of cells and tissues to the spatiotemporal induction of translation for biologically active molecules like morphogens.

### Local induction of protein expression at late developmental stages

To assess whether the method we present can be employed at late embryonic stages (beyond 24 h post-fertilization), we manipulated the migration of the posterior lateral line primordium (pLLP). The pLLP is comprised of a group of cells that migrate along the body of zebrafish embryos. This cohort of cells is directed by Cxcr4-Cxcl12a signaling, which is regulated by the expression of the receptor Cxcr7[32–34]. The directional migration of the pLLP depends on the self-generated Cxcl12a gradient produced by these cells, which relies on the expression of the scavenger receptor Cxcr7, specifically at the rear of the migrating primordium[32–34]. We, therefore, hypothesized that the induction of Cxcr7 expression at the front of the pLLP would disrupt the formation of the Cxcl12a gradient and decrease the migration distance of the pLLP. Indeed, when we measured the distance between the front of the pLLP and the end of the yolk extension at 48 h post-fertilization (L in Supplementary Fig. 15a), we observed a strong reduction in the migration of the primordium in irradiated embryos in which Cxcr7 expression was induced, compared to non-irradiated controls or irradiated embryos expressing a control protein (mCherry) (Supplementary Fig. 15b–e).

## Discussion

Optogenetic regulation of mRNA translation initiation is a useful tool in different experimental contexts, but the approaches available are not widely used. Some approaches rely on chemical modification of the transcript[6,7], which can impair the biological activity of the mRNA and are thus not readily applicable. Alternative approaches require using light-sensitive morpholino oligonucleotides (photo morpholinos), which must be designed for targeting specific genes of

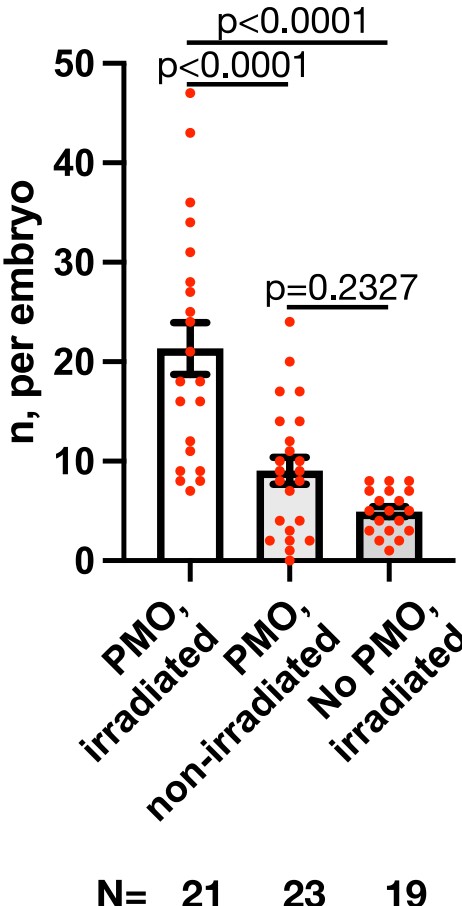

**Fig. 6 | Quantification of the efficiency of light-controlled induction of toxin expression.** The graph shows the number of apoptotic cells per embryo (n) based on active Cas-3 staining. The embryos were treated as described in Fig. 5. "PMO, irradiated" is equivalent to 'c' panel in Fig. 5. "PMO, non-irradiated" is equivalent to 'b' panel in Fig. 5. "No PMO, irradiated" is equivalent to 'a' panel in Fig. 5. N - number of embryos quantified, n- number of apoptotic cells per embryo. *P*-values were determined by a one-way ANOVA multiple comparisons test. Error bars represent SEM. Source data are provided as a Source Data file.

interest[10,11,13]. Due to sequence variability, synthesizing effective caged morpholinos can be challenging, and the efficiency of uncaging and biological activity cannot be guaranteed. The system we developed here overcomes these hurdles.

In this work, we have successfully developed a robust optogenetic system that uses a universal reporter construct with a tested functional 5' translation regulatory sequence. Translation of this reporter mRNA is blocked by the standard commercially available morpholino (Gene Tools), which is widely used and shows no off-target effects. Similarly, the sequence of the caged guanidinium-linked morpholino (GMO)-phosphorodiamidate morpholino oligonucleotide (PMO) chimera we use here is also constant, and it is complimentary to the standard translation-blocking morpholino. The semi-automated synthesis, caging, uncaging, and thus, the activity of such an oligonucleotide is easily standardized, making it suitable for widespread use. The metal-free method we developed for the synthesis of guanidinium linkage with the secondary amine has been established under mild conditions. Importantly, this approach could potentially be employed for the synthesis of various types of guanidines, which is relevant for a wide range of applications in natural products[35] and organocatalysis[36].

A difference between this method and a previously described system for inducing RNA translation[13] is that ours uses two

oligonucleotides rather than one. As such, our setup allows for more flexibility regarding the level of activation of the target RNA and the level of residual background translation. This feature is critical when expressing proteins of high specific activity (e.g., toxins).

Importantly, our method is not limited to use in developing zebrafish embryos and injection of the reagents. Rather, transfection of RNAs and morpholino antisense oligonucleotides can be performed into cells and organoids (reviewed in refs. 37–39). Furthermore, homologous recombination of the tested 5' sequence into the beginning of open reading frames of genes of interest and transfection of the morpholinos can also allow for modulating the translation of endogenous genes of interest.

Overall, we present an easy and broadly applicable system for light-induced and light-inhibited protein expression, thereby overcoming the main barriers to these types of technologies. This method relies on the development of light-activated morpholino oligonucleotide that allows for controlling protein levels, thereby facilitating specific biological effects to be exerted in the context of live embryos. As such, the chemical compound we synthesized and the strand-displacement approach will have a strong impact on basic research and medically relevant experiments.

## Methods
### Zebrafish husbandry and microinjections
Zebrafish maintenance was performed in compliance with the German, North-Rhine-Westphalia state law, following the regulations of the *Landesamt für Natur, Umwelt und Verbraucherschutz Nordrhein-Westfalen* and was supervised by the veterinarian office of the city of Muenster (Approval No: 53.5.32.7.1/MS-07825). All the regulations regarding ethical approval and the treatment of the animals and conditions in the facility are followed. The following zebrafish (*Danio rerio*) lines were employed: wild-type (WT) fish of the AB background, *Tg(kop:mcherry-F-nos3'UTR)*[40] and *Tg(gsc:GFP)*[41], *Tg(-8.0cldnB:lynGFP)*[32].

WT and manipulated embryos were collected, kept in 0.3× Danieau's solution [17.4 mM NaCl, 0.21 mM KCl, 0.12 mM MgSO$_4$·7H$_2$O, 0.18 mM Ca(NO$_3$)$_2$, and 1.5 mM HEPES (pH 7.6)], and raised at 28 °C.

For all experiments, embryos were microinjected into the yolk with 2 nl of mixtures containing the oligonucleotides and RNA molecules. Capped sense mRNAs were synthesized using the mMESSAGE mMACHINE (Thermo Fisher Scientific) following the manufacturer's protocol. Amounts of the injected mRNAs and concentrations of morpholinos used in this work is presented in Table 2. The amounts of the specific reporter mRNAs and photomorpholinos were calibrated for each PMO synthesis batch based on the biological activity of each reporter transcript. In the case of the tis-GFPnls mRNA injections, very weak nuclear GFP could be observed in 30% of the embryos before irradiation (defined as leakiness), and these were excluded from the experiments. The synthesized mRNAs included an SV40 3'untranslated region. *tis-kid* mRNA was synthesized from the PCR template as follows: The 1st PCR template (the ORF) was amplified from the plasmid (internal DB Nr 754) with primers K304 (5'-caTAAATTGTAAaTGAGGTAAGAGGgggatccaccatggaaagaggggaaatctggctt-3') / J168 (5'- tcccacacctccccctgaacctgaaa-3'). This fragment was used as the template to amplify the full sequence including the T3 promoter by the second PCR using primers K321 (5'-cccAATTAACCCTCACTAAAGgcaTAAATTGTAAaTGAGGTAAGAGG-3') / J168 (5'- tcccacacctccccctgaacctgaaa-3'). *tis-ctnnb1* mRNA was synthesized from the PCR template as follows: The 1st PCR template (the ORF) was amplified from the plasmid (internal DB Nr 405) with primers K517 (5'-TAAATTGTAAaTGAGGTAAGAGGaccatggctacccagtctgacttg-3') / K518 (5'-tcatgtctggatctacgtaatacgacttacagatcggtgtcaaacca-3'). This fragment was used as the template to amplify the full sequence, including the T3 promoter by the second PCR using primers K321 (5'-cccAATTAACCCTCACTAAAGgcaTAAATTGTAAaTGAGGTAAGAGG-3') / K518 5'-tcatgtctggatctacgtaatacgacttacagatcggtgtcaaacca-3'). The translation-blocking

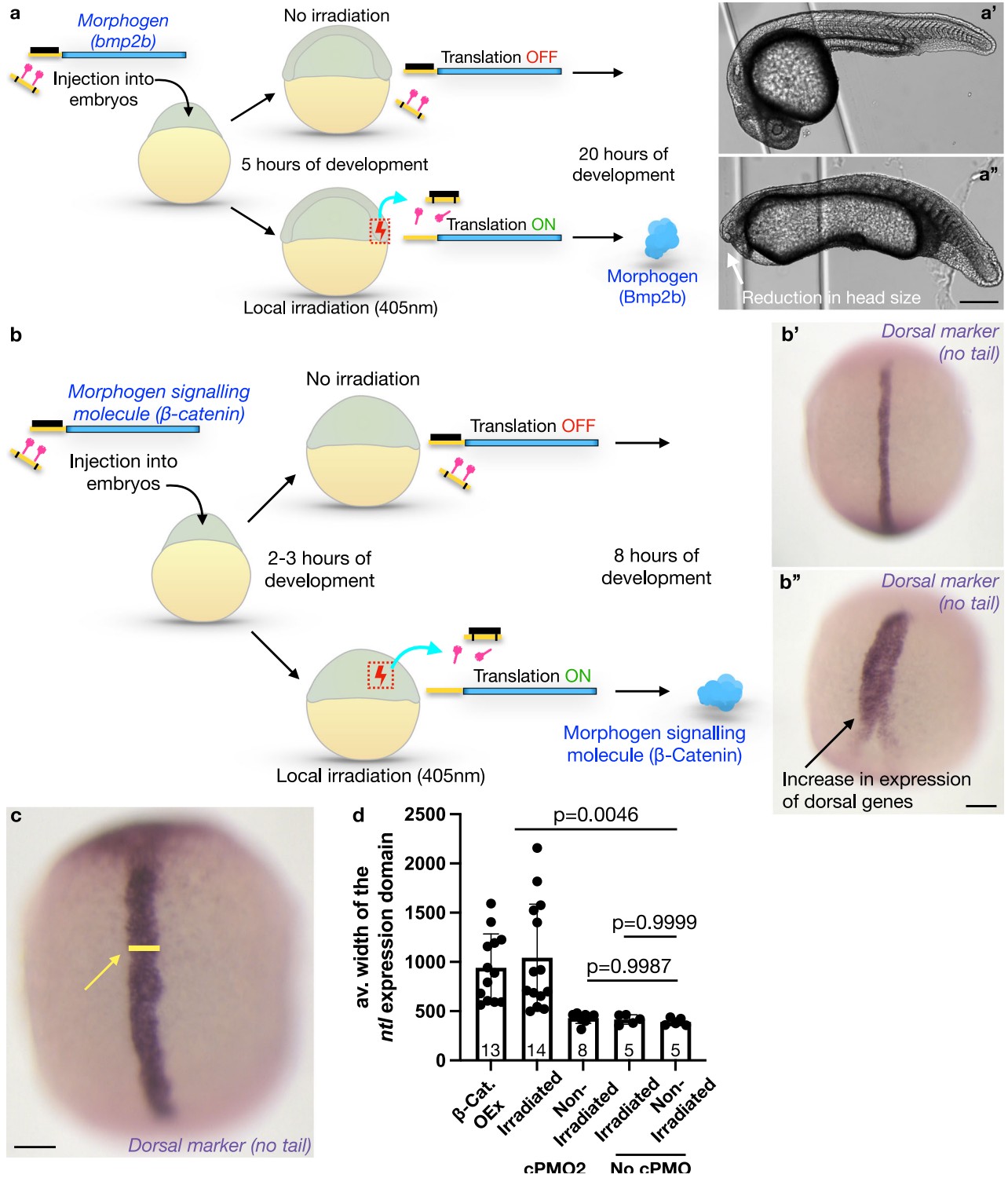

**Nature Communications** | (2025)16:3614

morpholino (tbMO, purchased from Gene Tools) utilized in this work is: 5'-CCTCTTACCTCAGTTACAATTTATA-3'. The nucleotide sequence located upstream of the ATG translation start codon is the translation initiation sequence complementary to the translation-blocking morpholino (yellow in all figures) 5'-TATAAATTGTAA**A**TGAGGTAAGAGG. The point mutation (C > A, bold in the sequence above) was introduced into the PMO sequence in the reporters used for the light-induced protein expression to facilitate dissociation of the translation-blocking morpholino from the reporter transcript and to ensure the most efficient sequestration of the translation-blocking morpholino by activated PMOs.

## Microscopy and PMO activation

For live imaging, embryos were dechorionated, transferred to 0.3× Danieau's solution, mounted in agarose-coated ramps covered with Danieau's solution, and manually oriented. Embryos older than 20 h post-fertilization (hpf) were anesthetized using tricaine (A5040, Sigma-Aldrich). Spinning disk confocal microscopy was performed using Carl Zeiss Axio Imager Z1 and M1 microscopes equipped with Yokogawa CSUX1FW-06P-01 spinning disk units. Imaging was performed using a Hamamatsu ORCA-Flash4.0 LT C11440 camera and Visitron Systems acquisition software (VisiView). Imaging of embryos older than 20 hpf was performed using the 5x objective, while younger

**Fig. 7 | Light-induced expression of morphogens affects zebrafish embryo development. a** Embryos were injected with mRNA encoding for a morphogen (Bmp2b) that inhibits the development of dorsoanterior structures, such as the head (blue), cPMO2 (yellow with black stripes and light-cleavable moieties in pink), and with the standard translation-blocking morpholino (black). Embryos were allowed to develop for 5 hours. Some of the embryos were raised in the dark (upper arrow), such that the translation of the mRNA was inhibited by the translation-blocking morpholino (**a′**). Sibling embryos were irradiated locally with the 405 nm laser at the region of the embryo that induces the development of dorsal structures such as the head (lower arrow, red box), resulting in photolysis of PMO2's light-sensitive moieties. This treatment led to the sequestration of the translation-blocking morpholino and the expression of the morphogen that inhibits head development (lower panels). The abnormal development of the head was documented in 25 h old embryos (**a″**, white arrow). **b, c** Embryos were injected with the mRNA encoding for the morphogen signaling molecule β-catenin, cPMO2 (yellow with black stripes with light-cleavable moieties (pink)), and with the standard translation-blocking morpholino (black). At 2 hpf a fraction of the embryos was raised in the dark (upper arrow), such that the translation of the mRNA was blocked by the translation-blocking morpholino, resulting in no morphogen signaling (**b′**). Sibling embryos were irradiated locally with the 405 nm laser (red dotted box, lower arrow), resulting in the photolysis of PMO2's light-sensitive moieties. Under these conditions, sequestration of the translation-blocking morpholino allowed the dorsalizing factor to be expressed (lower arrow). This treatment resulted in the strong expansion of dorsal structures, as detected by the expression of *no tail* (*ntl*) RNA, a dorsal gene marker (**b″**, black arrow and **c**, yellow line and arrow). **d** Quantification of this phenotype was performed by measuring the width of the *ntl* expression domain (**c**, yellow line, and yellow arrow). Scale bar 400 μm for **a′-a″**, 150 μm for **b′, c**. *P*-values were determined by a one-way ANOVA multiple comparisons test. Error bars represent SEM. See Supplementary Figs. 13, 14 for additional controls and quantifications. Source data are provided as a Source Data file.

**Table 2 | Amounts of the injected mRNAs and concentrations of morpholinos used in this work**

| RNA name | RNA, pg | cPMO, uM | tbMO, uM | Stage of irradiation, hpf | Irradiation, min |
|---|---|---|---|---|---|
| *tis-GFPnls* (internal DB Nr F041) | 20 | 500 | 25 | 4 | 5 |
| *tis-kid* (PCR template) | 5 | 500 | 50 | 10 | 7 |
| *tis-bmp2b* (internal DB Nr F045) | 10 | 200 | 30 | 4-5 | 9 |
| *tis-ctnnb1* (PCR template) | 100 | 500 | 10 | 2-3 | 7 |
| *tis-cxcr7* (internal DB Nr F374) | 600 | 500 | 25 | 24 | 3 |
| *tis-control* (internal DB Nr F374) | 300 | 500 | 25 | 24 | 3 |

embryos were imaged using the 10x objective. Image acquisition was conducted by acquiring 50–400 μm Z-stacks (Z planes 5 μm apart).

Laser irradiation for the activation of the photomorpholino was performed using the 405 nm laser (34 mW) of 100% laser power for 3–9 min (see Table 2) under 63x objective.

Confocal laser scanning image acquisition was performed using an LSM710 (Zeiss) upright microscope and ZEN software (Zeiss). Imaging was performed using a 20x water-dipping objective with 2 μm Z optical slices.

### Transplantations
Donor embryos were generated by co-injection at the 1-cell stage of 500 pg of Dextran conjugate (Invitrogen™, Alexa Fluor™ 680; 10,000 MW, Anionic, Fixable, Catalog number D34680) and one of the three different 10 μM BODIPY-488-labeled PMOs (PMO1, PMO2 or PMO3). Embryos injected with 80 pg of *mCherry-F′-globin* mRNA (labeling all cell membranes) served as hosts. At 4 hpf, 20–30 donor cells were transplanted into host embryos of the same stage. At 5-6 hpf, transplanted embryos were subjected to confocal imaging.

### Immunohistochemistry
Whole-mount in situ hybridization using the Digoxigenin (DIG)-labeled probe for *no tail* (*ntl*) RNA was performed as previously described[42].

Whole-mount immunostaining for the detection of activated caspase-3 was performed as described previously[25,24]. The primary antibody (Purified Rabbit Anti-Active Caspase-3, Clone C92-605- BD Biosciences) was used at a 1:500 dilution. The secondary antibody (Goat anti-Rabbit Alexa Fluor 647, Thermo Fisher Catalog number A-21245; RRID: AB_2535813) was employed in a 1:1000 dilution. Nuclei were visualized by incubating the embryos in 0.002 mM Hoechst solution (Thermo Fisher) overnight at 4 °C.

### Statistical analysis
Statistical analysis (Student's *t* test or ANOVA) was performed using GraphPad Prism software (version 8). All biological experiments were performed in three independent replicates. No data was excluded from the analysis. Experiments were not blinded.

### Electrophoretic mobility shift assay
Electrophoretic mobility shift assays were performed using Biorad Power Pac Basic at 25 °C at 120 V for 1.5 h in TBE (Tris borate EDTA 1x) buffer on a native polyacrylamide gel (20%). The duplexes were formed by mixing of the tbMO and RNA strands in an equimolar ratio to obtain a 100 μM stock solution in PBS 1X buffer (137 mM NaCl, 2.7 mM KCl, 10 mM $Na_2HPO_4$, and 1.8 mM $KH_2PO_4$). The mixture was then annealed in a thermocycler (S1000 Thermal Cycler, Bio-Rad) from 95 °C for 5 min and then cooled to 4 °C in 10 mins. After that, the duplex was stored at 4 °C for 12 h. After that, RNA and duplexes were incubated with strand displacing PMO (PMO1: 3 equivalents or 6 equivalents and PMO2: 6 equivalents) in annealing buffer (PBS 1X) at 30 °C for 3 h, 6 h and 12 h in 50 μM final concentration. The samples were loaded using TBE native dye (5x, 50% glycerol, 0.05% bromophenol blue). The RNA and duplexes were stained by Ethidium Bromide and visualized by scanning and image capture using Biorad ChemiDoc MT imaging. Images were analyzed using ImageJ software, and data were plotted using GraphPad Prism 6.0 software.

### Thermal denaturation studies
Thermal melting spectra were recorded at (Carry 3500 UV-Vis Peltier Spectrophotometer) using quartz cuvettes with a 1 cm path length. Melting temperatures were recorded at 260 nm with a ramp of 1 °C/ min. The duplex samples were obtained by mixing both strands of interest in a 1:1 ratio at 2 μM concentration in 40 mM phosphate buffer followed by heat to 85 °C for 5 min, then with slow cooling to temperature ranges from 85 °C to 10 °C with an interval of 1 °C/min. Origin 9.1 was used to determine all $T_m$ values from the first derivative curves.

### CD-spectral experiments
All PMO-RNA and PMO-PMO Circular Dichroism experiments were performed using 2 μM concentrations of each strand and 0.04 M phosphate buffer in a JASCO J-1500 Spectropolarimeter. Scanning rate 200 nm/min and bandwidth 1 nm and data accumulation 3 times. All the samples were allowed to anneal at 85 °C for 5 min and then cooled slowly to 10 °C at the rate of 1 °C min⁻¹. Then all the duplexes were stored at 4 °C. All data collection was carried out at 10 °C.

## Flow cytometric experiments

HeLa cells were grown in 12-well plates, grown to 50% confluency for 24 h (37 °C, 5% $CO_2$). Compound incubation for cellular uptake measurements was done for 4 h. Flow cytometry studies were performed on a BD FACS Aria III and a BD LSR Fortress and analyzed using the FlowJo software. Mean comparisons were analyzed by Graph Pad Prism 6.

## Live cell imaging

HeLa cells were seeded in 12 well plate and grown to 50% confluency for 24 h (37 °C, 5% $CO_2$). Compound incubation for uptake was done for 24 h, and the cells were imaged live after a PBS wash. Imaging was performed on Olympus IX51 microscope, image processing was done using Fiji (ImageJ).

## General chemical methods

All chemical reagents were purchased from commercially available sources and used without further purification unless otherwise specified. Reactions were conducted in glassware that had been thoroughly dried in the oven and placed under an argon atmosphere. Solvents were purified and dried following standard protocols. Thin layer chromatography (TLC) was performed using silica gel 60 F254-coated aluminum sheets (0.25 mm layer thickness, Merck). TLC visualization was accomplished using UV light (254 nm), and staining was carried out using standard staining solutions such as CAM, Ninhydrin, etc. For purification, column chromatography has been performed on silica gel columns (mesh sizes: 60–120 and 100–200). Nuclear Magnetic Resonance (NMR) spectra, including $^1H$, $^{13}C\{^1H\}$, and $^{31}P$ NMR, were recorded using Bruker NMR spectrometers operating at 300 MHz. $^{13}C\{^1H\}$ NMR spectra were recorded at 75 MHz. $^{31}P$ NMR spectra were recorded at 121 MHz. Chemical shifts ($\delta$) are reported in parts per million (ppm) relative to the solvent residual peak ($\delta = 7.26$ ppm, for $CDCl_3$) or the TMS standard. The multiplicity of NMR signals is abbreviated as follows: singlet (s), doublet (d), triplet (t), septet (sept), broad signal (brs), or multiplet (m). High-resolution Mass Spectra (HRMS) were obtained using a QTOF I (Quadrupole hexapole TOF) mass spectrometer equipped with an orthogonal Z spray electrospray interface on a Micro (YA 263) mass spectrometer (Manchester, UK). Matrix-Assisted Laser Desorption Ionization (MALDI) mass spectra were recorded using a Bruker UltrafleXtreme MALDI-TOF/TOF system. The matrix used was Sinapinic acid (SA). High-Performance Liquid Chromatography (HPLC) purification of all the PMOs were performed using a Shimadzu SPD-20A system and $C_{18}$ (Waters XBridge BEH Shield RP18) column, employing a gradient system: 20–50 % MeCN/ 0.1 M ammonium acetate (in water, pH 7.12) for 20 min and then 50-20% MeCN in 0.1 M ammonium acetate for 10 min.

## Synthesis and purification of the photomorpholinos

These procedures are presented in Supplemental Information (Supplementary Figs. 16–29)

## Reporting summary

Further information on research design is available in the Nature Portfolio Reporting Summary linked to this article.

## Data availability

No large-scale datasets have been generated in this study. This paper does not report the original code. All plasmids generated in this study are available from the lead contact (erez.raz@uni-muenster.de). Information or requests for biological and chemical resources and reagents should be directed to Erez Raz (erez.raz@uni-muenster.de), and Surajit Sinha (ocss5@iacs.res.in) respectively. Source data are provided in this paper.

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

## Acknowledgements

This work was supported by the *Deutsche Forschungsgemeinschaft* (DFG) (SFB 1348 (B06) and RA 863/17-1 for E.R.) and the Medical Faculty of the University of Münster (E.R. and K.T.). S.S. thanks Technical Research Center (TRC) and DST, New Delhi, Government of India (DST/TDT/TC/RARE/2022/10c2) for DNA synthesizer and funding support. We thank Celeste Brennecka for the critical reading of the manuscript and Laura Ermlich for help with the graphical design. We also acknowledge Ines Sandbote, Esther-Maria Messerschmidt, and Ursula Jordan for excellent technical assistance.

## Author contributions

E.R. and S.S. conceived the idea, designed the hypothesis, and supervised the project. K.T. performed all the zebrafish experiments and analyzed the data. A.G. and A.D. synthesize all the PMO oligonucleotides. A.G. conducted all the chemistry experiments and analyzed the data. D.K. optimized the mercury-free GMO synthesis. S.N.S. performed in vitro experiments and analyzed the data. All authors participated in writing the manuscript and approved the final version.

## Funding

## Competing interests

The authors declare no competing interests.
