## [Transparent Peer Review file · Nature Communications]

Optochemical control over mRNA translation by photocaged phosphorodiamidate morpholino oligonucleotides in vivo

Corresponding Author: Professor Erez Raz

Version 0:

Reviewer comments:

Reviewer #1

(Remarks to the Author)

In this paper, the authors report a new method for conditional gene activation from ectopically delivered synthetic mRNA. They use this in an in vivo animal model, the zebrafish. The authors are expert developmental biologists, and they use this novel method in activating fluorescent proteins, morphogens, and cell death proteins. The method works through leveraging the extremely effective use of morpholinos to inhibit mRNA translation through a morpholino, that is then reversed by a caged / photomorpholino. The latter is activated by light and then removes the translational block. As presented, the data looks consistent and reproducible. The chemistry that has been used is innovative in its synthesis, including some of their molecules able to be made on a commercial system. The use of a single sequence for all exogenous mRNA molecules means this method can be readily adaptable to use by nearly every all mRNAs. This tool will likely be applicable for other systems such as *Xenopus* and *Drosophila*.

Major question:

What is the range of the outcomes they see? They report that each experiment was conducted in triplicate. But how many embryos were assessed each time, what were the range of effects they see? Did they see ANY leaky mRNA injections? What about mosaic effects (if any)?

Reviewer #2

(Remarks to the Author)

Tarbashevich et al. present a novel method to optically control translation of exogenously added mRNAs with a combination of a translation-blocking antisense morpholino oligonucleotide and a complementary caged guanidinium-phosphordiamidate (GMO-PMO) chimeric oligonucleotide. The authors present schemes for chemical synthesis of the guanidinium-linked and caged portions of the oligos and develop a mild synthesis strategy that makes the guanidinium-linked portion compatible with semiautomated oligo synthesizers for synthesis of the phosphordiamidate portion. They demonstrate the utility of the system in zebrafish embryos with induction of broad GFP expression, local induction of a specific dorsal patterning morphogen, local induction of a toxic protein, and with local inhibition of translation of a transcript. On the whole, this is a very useful technique that will provide the ability to induce expression of added engineered mRNAs in a spatially localized manner. The research is well done and the manuscript generally well presented. I do have two concerns with the experiments and have commented on a number of places in the manuscript that are unclear or may need revision. As a general comment, the figures in the main text have avoided any quantification of effects of the system on gene expression/embryo development, leaving any quantitative measures in the supplements. This is undesirable and reduces confidence when reading the study.

Significant problems:

1. The study does not rule out an effect of the caging molecules on cells after the cPMO is uncaged. The induction of GFP expression in figure 3 implies that gene expression can continue, but a quantitative assessment would be helpful. The translation-blocking experiment in figure 6 and S10 could provide this control, as injection of a non-targeting cPMO would allow you to separate the effect of the caging molecules from the effect of the translation-blocking PMO on inhibition of GFP expression.

2. The electrophoretic mobility shift experiment in figure S6 is important to show that there is binding of the PMOs to the target RNA. However, the experiment and gel are extremely difficult to interpret. Why would single-stranded RNA be degraded by incubation, but the entire RNA protected when complexed with the MO? This is an in vitro incubation, isn't it? Something is wrong with this experiment, and a better experimental design should be used. There should not be any need to incubate overnight; binding should happen in minutes or seconds at an appropriate temperature. When improved, please provide more detail in the methods. Composition of the annealing buffer is needed; the method of labeling the RNA with FAM is needed.

Minor problems that detract from readability and interpretability:

1. Results page 3: The first paragraph of results is confusing regarding what is included in the system. The paragraph states that the system includes an antisense morpholino oligo targeting a specific RNA sequence and a sense caged PMO "which is complimentary to a standard negative control Morpholino." What is meant by "which is complimentary to a standard negative control Morpholino"? Isn't this complementary to your targeting antisense MO? The methods do not help to clarify this statement, simply stating that this was purchased from GeneTools.

2. Results page 3: Some solvents are mentioned as abbreviations that are never defined. Please define ACN/DCM. Several others are also mentioned just as abbreviations but are defined elsewhere in the manuscript. Searching for the meaning of these abbreviations is distracting to the reader, but at least the confusion can be resolved for the other abbreviated solvents and chemical constituents used. A more systematic naming of abbreviated terms at their first appearance would be helpful.

3. Scheme 1: The legend of scheme 1 labels NEM as N-methylmorpholine, while all other mentions of this compound indicate N-ethylmorpholine.

4. Results: Table I: Duplexes of what? The table legend needs to indicate what is duplexed with each oligo/RNA (is it PMO4?).

This table disagrees with figure S3. Figure S3 has the RNA-PMO4 duplex with $T_m = 36$ and PMO4-PMO3 = 48, while this table indicates RNA $T_m = 48$ and PMO1 = 36. In fact, all T_m values disagree except cPMO2 before irradiation.

5. Results, page 8: Is PMO4 the standard negative control morpholino, or are you co-injecting PMO4 with the standard negative control morpholino?

6. Results figure 3: If this experiment was performed with local uncaging, please show a low-power image of the embryo revealing the extent of activation of expression.

Why are you showing GFP expression in blue? This color is difficult for readers to see.

7. Results, page 9: It was stated that there was reduced efficiency inducing GFP expression with cPMO1 compared to cPMO2, with a reference to the in vitro gel shift data in figure S6. As mentioned before, figure S6 is difficult to interpret as it is. It would be preferable to see the GFP induction data in figure 3, along with quantification of the induction by uncaging cPMO2 and cPMO1.

8. Results Figure 4: The caspase-3 labeling and its quantification should be shown in the main figure 4, not in a supplement.

9. Results Figure 5 morphogens: We need to see some quantification of the effects of local release of bmp2b. The image of one mis-developed embryo is compelling, but how often was this observed? Same comment for release of B-catenin. Please quantify something and report the outcome.

10. Figure S9: Is the figure in b correct? It indicates that the injected RNA expresses B-catenin, but the legend indicates it is the same bmp2b as in part a. The figure also does not include the caged PMO, although text in the main document (page 12) indicates that it is included.

11. Results Figure 6 and S10. Significantly more information is needed about the GFP reporter used for these studies. How does this reporter direct GFP expression to a limited subset of cells? The authors refer to a review article, which indeed suggests that there are sequences that restrict certain maternal RNAs to express in primordial germ cells, but being a review, does not give any technical information. This is not acceptable. The methods need to provide this information and the results should be more clear about what is being used.

Reviewer #3

(Remarks to the Author)

The manuscript "Optochemical control over mRNA translation by photocaged phosphorodiamidate morpholino oligonucleotides in vivo" describes the use of photocaged guanidinium-linked morpholino (GMO)- phosphorodiamidate morpholino oligonucleotide (PMO) chimaeras to control protein translation. The core concept is based on the injection of two probes, which reduce synthetic economy as well as increase both complexity and potential side effects. Under irradiation, one is uncaged, displacing the one that blocks mRNA initiation translation site, thereby triggering protein translation. While the manuscript is comprehensive and includes exhaustive interesting biological data, particularly demonstrating the spatiotemporal control of translation, there are several concerns that limit its impact, especially for a high-profile journal like Nature Communications.

First, the term "in vivo" may be somewhat misleading as the targeting appears to be "ex vivo" and dependent on in vitro blockage that must be injected. This raises questions about the broader applicability of the technique ("universal technique"?), as the authors claim.

Second, although the study introduces a never-reported approach, I struggle to see any significant advantage of using this methodology. On the contrary, if we compared this strategy with more straightforward techniques such as inducible promoters or photocleavable/photocaged siRNA, miRNA, PMOs, and PNAs. In particular, the use of GMO-PMO conjugates adds unnecessary synthetic complexity compared to traditional photoprotected PMOs or other oligo-probes. The authors claim benefits from the GMO modification, but no compelling evidences are provided to support this. For example, while it is suggested that the GMO increases uptake, the probes are injected, and no data is shown to demonstrate improved cell permeability. Along these lines, the authors need to demonstrate that the GMO modification increases (i) the affinity towards unmodified complementary PMO (as their data suggests a 2°C reduction compared to PMO alone, which indicates indeed reduced affinity), (ii) benefits the on/off kinetics of strand displacement and (iii) significantly improve cell-uptake in a human cell line model for example HeLapLuc705.

Furthermore, the reduced affinity between PMO and RNA, compared to PMO/PMO-GMO hybrids, forces an excess of PMO, which diminishes the applicability of this method. Even with this excess of PMO/GMO, a significant portion of RNA/PMO duplexes remains intact, which limits the system's efficiency.

In summary, while the data presented may be interesting and potentially publishable, the overall impact of this study does not meet the high standards expected for a journal like Nature Communications. I recommend redirecting this manuscript to a different journal more suited to the impact of this contribution, and suggest the authors provide a clearer justification for the use of such complex conjugates, supported by objective demonstrations of their benefits compared to simpler, existing technologies. The precedents in the field should also be included in the new manuscript version.

Version 1:

Reviewer comments:

Reviewer #1

(Remarks to the Author)

The authors have done an excellent job addressing the key comments from my reviews. It's ready to be shared so that the true value (good and limits) of the technology will be identified based on different labs using this in diverse application areas.

Reviewer #2

(Remarks to the Author)

The revised manuscript is significantly improved. The authors have clarified the questions I have posed and made terminology within the manuscript consistent and easier for readers to follow. Furthermore, addition of quantitative measures of effects is important. I believe that this novel technique adds a new tool to study developmental biology questions, allowing controlled regional induction of protein expression.

Reviewer #3

(Remarks to the Author)

The authors have addressed the concerns. Therefore, if the editor finds the impact of this work suitable for Nature Communications, I have no objections. However, before publication, please correct the statement regarding the precedent <https://doi.org/10.1038/s41557-022-00972-7>, as it has also been used in zebrafish <https://doi.org/10.1038/s42004-025-01411-7>.

We thank the referees for their constructive feedback, which we have addressed and incorporated into the revised version of the manuscript. Below, we present our responses to the comments and details of the new experiments conducted.

Before addressing the specific issues raised, we would like to respond to a general point made by Referee 3 regarding the motivation behind our work and how it compares to existing methods for light-induced protein expression.

The referee questions the novelty and utility of our method in light of alternatives that are described as “more straightforward techniques such as inducible promoters or photocleavable/photocaged siRNA, miRNA, PMOs.” Methods for induction of protein expression that rely on the initiation of RNA transcription are indeed powerful and extensively used but typically involve the generation of stable transgenic animals for each gene of interest, which in practice means not less than 6-7 months before one can work with the line (John Collin and Paul Martin – Chapter 13 in Basic Science Methods for Clinical Researchers, 2017, <http://dx.doi.org/10.1016/B978-0-12-803077-6.00013-8>). To further illustrate the system's robustness and demonstrate that it can operate effectively at more advanced stages of development without generating transgenic animals, we conducted a new experiment shown in Figure S12. In addition to the long time required for generating transgenic animals, in principle, activation of translation rather than transcription should provide a more rapid accumulation of protein.

Thus, to obtain rapid induction of protein expression, one must utilize RNA-translation-induction-based methods. To this end, photocleavable photomorpholinos were developed (Tallafuss et al. 2012, DOI: 10.1242/dev.072702); however, beyond the original publication, this method has not been employed for the induction of gene expression *in vivo*. These limitations have motivated efforts to induce translation using a modified 5' RNA Cap, which facilitates translation upon light irradiation (see Klöcker et al. 2022, DOI: 10.1038/s41557-022-00972-7). This method, recently published in Nature Chemistry, appears in principle effective in cell culture, but expression of the induced proteins required over 24 hours. This system has not been shown to function in biological experiments, particularly not in the context of live developing embryos as here.

The method we present is novel, allowing for a robust and flexible application of light-modified compounds within the more challenging model of cells located deep in a developing embryo. In addition, the underlying principle- strand displacement via an activated photomorpholino- is original and holds potential for future applications as other species of photomorpholinos are developed. The system's robustness, as demonstrated with five different RNA molecules, can thus be scaled up for various purposes, including, for example, the screening for immediate biological responses in cells or embryos.

Reviewer #1 (Remarks to the Author)

> In this paper, the authors report a new method for conditional gene activation from ectopically delivered synthetic mRNA. They use this in an *in vivo* animal model, the zebrafish. The authors are expert developmental biologists, and they use this novel method in activating fluorescent proteins, morphogens, and cell death proteins. The method works through leveraging the extremely effective use of morpholinos to inhibit mRNA translation through a morpholino, that is then reversed by a caged / photomorpholino. The latter is activated by light and then removes the translational block. As presented, the data looks consistent and reproducible. The chemistry that has been used is innovative in its synthesis, including some of their molecules able to be made on a commercial system. The use of a single sequence for all exogenous mRNA molecules means this method can be readily adaptable to use by nearly every all mRNAs. This tool will likely be applicable for other systems such as *Xenopus* and *Drosophila*.

We thank the referee for the positive view of the paper and for the suggestions for improving it.

> Major question:

> What is the range of the outcomes they see? They report that each experiment was conducted in triplicate. But how many embryos were assessed each time, what were the range of effects they see? Did they see ANY leaky mRNA injections? What about mosaic effects (if any)?

We thank the referee for pointing it out and have now included this information within the figures and the text.

The level of inhibition of RNA translation by the “standard GeneTools morpholino/blocking morpholino” depends on the concentrations of the RNA and the morpholino. For example, when inducing the expression of a fluorescent protein, one typically injects relatively high amounts of RNA to ensure sufficient protein levels for signal detection (20 pg for the nls-GFP encoding RNA). Conversely, when the protein exerts its function at very low amounts, one injects less RNA and more of the morpholino (5 pg for the Kid encoding RNA). We consider the ability to modify these ratios an important and unique feature of the new method.

In the case of the BMP and Kid-encoding RNAs, we detected no leakiness, as the “blocking morpholino” was present in a very high molar ratio relative to the RNA. To address this point, we have generated a new Figure (S10), where we present the range of phenotypes observed. In the case of GFP expression, where the amount of RNA is higher, very weak nuclear GFP was observed in 30% of the embryos before irradiation (which we define as leakiness), and these were subsequently excluded from the experiment and not irradiated to induce GFP expression. We have included this information in the methods section. Additionally, at the request of Referee 2, we have added a “zoom-out” of the results in Figure S6.

All of the experiments were conducted in triplicate, and we now provide the total number of embryos in the figure legend, along with the precise number of embryos for each repetition in a new Table (S3).

Reviewer #2 (Remarks to the Author)

> Tarbashevich et al. present a novel method to optically control translation of exogenously added mRNAs with a combination of a translation-blocking antisense morpholino oligonucleotide and a complementary caged guanidinium-phosphordiamidate (GMO-PMO) chimeric oligonucleotide. The authors present schemes for chemical synthesis of the guanidinium-linked and caged portions of the oligos and develop a mild synthesis strategy that makes the guanidinium-linked portion compatible with semiautomated oligo synthesizers for synthesis of the phosphordiamidate portion. They demonstrate the utility of the system in zebrafish embryos with induction of broad GFP expression, local induction of a specific dorsal patterning morphogen, local induction of a toxic protein, and with local inhibition of translation of a transcript. On the whole, this is a very useful technique that will provide the ability to induce expression of added engineered mRNAs in a spatially localized manner. The research is well done and the manuscript generally well presented. I do have two concerns with the experiments and have commented on a

number of places in the manuscript that are unclear or may need revision. As a general comment, the figures in the main text have avoided any quantification of effects of the system on gene expression/embryo development, leaving any quantitative measures in the supplements. This is undesirable and reduces confidence when reading the study.

We thank the referee for the very useful comments and for pointing out the lack of quantitative information in the original version of the manuscript. As outlined in the specific responses, this information has now been incorporated into the manuscript.

Significant problems:

1. The study does not rule out an effect of the caging molecules on cells after the cPMO is uncaged. The induction of GFP expression in figure 3 implies that gene expression can continue, but a quantitative assessment would be helpful. The translation-blocking experiment in figure 6 and S10 could provide this control, as injection of a non-targeting cPMO would allow you to separate the effect of the caging molecules from the effect of the translation-blocking PMO on inhibition of GFP expression.

We understood that Referee 3 considered the biological experiments too complicated/numerous. The possibility of inhibiting RNA translation was a side point, which does not involve the new approach of strand displacement and in principle, inhibition of translation was employed before using other types of light-controlled oligonucleotides. We thus tried to simplify the paper by focusing only on the activation of translation and the “strand displacement” approach, which are the novel parts. We thus omitted the more complicated translation-blocking experiment that involved GFP signal bleaching and quantification of differences in recovery rate (Figure 6, numbering in the original version, see point 11). Instead, we strengthen the induction of translation by demonstrating it for a later process (Figure S12). Together with the request to add the quantitation of the data presented in Figure 4 (induction of cell death) to the main part, we now have the same number of main figures (the quantitation could not fit Figure 4).

As we understand this concern, it was not clear if, upon irradiation of cPMO, the caging moieties released are harmful to the cells. This point is indeed very important and was not explained/addressed in the original version of the paper. As stated by the Referee, the data in Figure 3 implies that translation can occur in the presence of the released caging moieties. However, a concern would be indeed that the released moieties affect the expression of other non-target genes (e.g., inhibit transcription). We believe that this is not the case for the following reason - Artificially increasing the Wnt morphogen signaling by induction of the downstream β -Catenin expression upon cPMO2 uncaging results in an expansion in the domain where transcription of the dorsal RNA marker (*no tail* RNA) occurs (Figure 6 in the current version). This effect is now quantified and presented (Figure 6 b-d). Hence, photolysis of the caging moieties does not affect the expected reaction to the manipulation as manifested by the activation of promoters regulated by β -Catenin. We now refer to this point in the main manuscript text. The lack of effect of the released moieties can also be appreciated from the data in Figure S12. There, we induced the expression of the decoy receptor *Cxcr7* in the lateral line. Following this treatment, the expression of GFP is maintained - compare Figure S12 panel c (non-irradiated) with e (irradiated, with biological effect).

2. The electrophoretic mobility shift experiment in figure S6 is important to show that there is binding of the PMOs to the target RNA. However, the experiment and gel are

extremely difficult to interpret. Why would single-stranded RNA be degraded by incubation, but the entire RNA protected when complexed with the MO? This is an in vitro incubation, isn't it? Something is wrong with this experiment, and a better experimental design should be used. There should not be any need to incubate overnight; binding should happen in minutes or seconds at an appropriate temperature. When improved, please provide more detail in the methods. Composition of the annealing buffer is needed; the method of labeling the RNA with FAM is needed.

Based on the *in vivo* experiments, the interaction between the translation-blocking morpholino and the RNA is immediate (or even occurring in the tube before injection) since, when titrated properly, there is no leakage manifested by protein production. E.g., expression of the toxin would lead to an extremely fast death of the embryo (Labbaf et al., 2022 <https://doi.org/10.1016/j.devcel.2022.07.008>). Based on the cell death observed by microscopy, apoptosis is observed within an hour after the uncaging of the PMO, so the initiation of protein expression starts much earlier. Thus, according to the *in vivo* data, all the interactions and release of the RNA occur rapidly.

We have repeated the electrophoretic mobility shift assay at different time points (Figure S6 in the current version). Conducting the experiment with reduced incubation time (3h, 6h, and 12h) and detecting the RNA by Ethidium bromide staining (instead of 5'-RNA-labeling by 6-FAM phosphoramidite on solid support), we now observed that the single-strand RNA remained intact throughout the experiment.

The experiment was conducted in PBS 1X buffer with strand displacing PMO (PMO1 and PMO2) in a thermal cycler maintained at 30°C. This information is now provided in the methods section.

Minor problems that detract from readability and interpretability:

> 1. Results page 3: The first paragraph of results is confusing regarding what is included in the system. The paragraph states that the system includes an antisense morpholino oligo targeting a specific RNA sequence and a sense caged PMO; which is complimentary to a standard negative control Morpholino. What is meant by "which is complimentary to a standard negative control Morpholino"? this complementary to your targeting antisense MO? The methods do not help to clarify this statement, simply stating that this was purchased from GeneTools.

The referee understood it correctly, but the phrasing in the original version was indeed not clear, and we have made an effort to clarify it as follows:

" The protein expression strategy we established relies on strand displacement of the target mRNA from a complex with the translation-blocking morpholino (tbMO). The displacement of the RNA occurs by an uncaged photomorpholino (PMO). The system thus comprises a standard Morpholino (Gene Tools, hereafter referred to as tbMO, indicated by the black bar at the bottom left cartoon in Figure 1) that targets a specific RNA sequence (represented by the yellow bar in Figure 1), and a photocaged PMO (cPMO, depicted by the yellow striped bar in Figure 1). Uncaging the PMO that has a sequence complementary to the tbMO leads to binding of the two and release of the RNA for translation."

> 2. Results page 3: Some solvents are mentioned as abbreviations that are never defined. Please define ACN/DCM. Several others are also mentioned just as abbreviations but are defined elsewhere in the manuscript. Searching for the meaning

of these abbreviations is distracting to the reader, but at least the confusion can be resolved for the other abbreviated solvents and chemical constituents used. A more systematic naming of abbreviated terms at their first appearance would be helpful.

As requested, we define the abbreviations the first time they appear in the text.

> 3. Scheme 1: The legend of scheme 1 labels NEM as N-methylmorpholine, while all other mentions of this compound indicate N-ethylmorpholine.

We apologize for the mistake, which we have corrected in the revised manuscript.

***> 4. Results: Table I: Duplexes of what? The table legend needs to indicate what is duplexed with each oligo/RNA (is it PMO4?).
> This table disagrees with figure S3. Figure S3 has the RNA-PMO4 duplex with $T_m = 36$ and $PMO4-PMO3 = 48$, while this table indicates RNA $T_m = 48$ and $PMO1 = 36$. In fact, all T_m values disagree except cPMO2 before irradiation.***

We thank the Referee for noticing these discrepancies, which were indeed mistakes corrected in the current version. We corrected Table I, showing the duplex composition. We have also provided the table legend and specified the thermal melting conditions and complementary PMO sequence.

> 5. Results, page 8: Is PMO4 the standard negative control morpholino, or are you co-injecting PMO4 with the standard negative control morpholino?

The two labs indeed used different terminologies for the same reagent. The term PMO4 refers to the commercially available "translation-blocking morpholino" from Gene Tools (tbMO). We have revised the figures and text to ensure consistency in the definitions throughout the manuscript.

> 6. Results figure 3: If this experiment was performed with local uncaging, please show a low-power image of the embryo revealing the extent of activation of expression.

As requested, we now provide a new Figure (S6) with "zoomed out" images of embryos subjected to treatments such as those presented in Figure 3. At low magnification, the GFP signal does not appear very bright after 2 hours (the time used in Figure 3). Therefore, we let the embryos develop for three more hours, leading to more dispersion of the cells.

> Why are you showing GFP expression in blue? This color is difficult for readers to see.

As requested, we have changed the blue color to green in Figure 3.

> 7. Results, page 9: It was stated that there was reduced efficiency inducing GFP expression with cPMO1 compared to cPMO2, with a reference to the in vitro gel shift data in figure S6. As mentioned before, figure S6 is difficult to interpret as it is. It would be preferable to see the GFP induction data in figure 3, along with quantification of the induction by uncaging cPMO2 and cPMO1.

Indeed, the data about this point was not included in the original submitted paper. We have observed that compared to cPMO1, the photoactivation and induction of GFP expression were notably more effective when cPMO2 was employed. We now state in the text that, for cPMO2, 89% of the injected embryos in which local expression was induced exhibited GFP expression. In contrast, only 45% of embryos treated with cPMO1 showed a GFP signal in the irradiated area.

> 8. Results Figure 4: The caspase-3 labeling and its quantification should be shown in the main figure 4, not in a supplement.

As requested, we have moved the caspase-3 labeling results to the main figures. The primary results are shown in Figure 4 and due to size constraints, the quantitation is presented in a new figure (Figure 5) in a graph.

> 9. Results Figure 5 morphogens: We need to see some quantification of the effects of local release of bmp2b. The image of one mis-developed embryo is compelling, but how often was this observed? Same comment for release of B-catenin. Please quantify something and report the outcome.

This information was missing in the original version and we provide it now. As is commonly used for such phenotypes we grouped the embryos according to the degree of severity (see, for example, Figure 1 in <https://pmc.ncbi.nlm.nih.gov/articles/PMC11472394/>). The results are illustrated in the new Figure S10. Our findings indicate that 80% of the embryos exhibit ventralisation upon uncaging and local activation of BMP2 expression in the dorsal organizer. 50% of the treated embryos exhibited what is defined as “severe ventralization”. For β -Catenin, the quantification is based on the width of the dorsal axial structures labeled by the induced expression of *ntl* RNA. This quantification is now presented in a new Figure 6 (c-d).

> 10. Figure S9: Is the figure in b correct? It indicates that the injected RNA expresses B-catenin, but the legend indicates it is the same bmp2b as in part a. The figure also does not include the caged PMO, although text in the main document (page 12) indicates that it is included.

We apologize for this copy-and-paste error and thank the referee for noticing it. We have corrected the figure legend and the text accordingly (Figure S11 in the current version).

> 11. Results Figure 6 and S10. Significantly more information is needed about the GFP reporter used for these studies. How does this reporter direct GFP expression to a limited subset of cells? The authors refer to a review article, which indeed suggests that there are sequences that restrict certain maternal RNAs to express in primordial germ cells, but being a review, does not give any technical information. This is not acceptable. The methods need to provide this information and the results should be more clear about what is being used.

In the original version, we have tried to make the paper more accessible to non-developmental biologists, hoping people from other fields are encouraged to employ the system. Thus, we did not include many details that are indeed required to fully understand this specific experiment. As also indicated by Referee 3, the biological experiments were difficult to understand. Since the strand-displacement approach and the activation (rather than inhibition) are the most important points of the paper, we have omitted the complicated inhibition experiment in the new version.

The explanation for the expression in the germ cells is as follows- GFP was expressed preferentially in germ cells based on the stabilization of RNA and reduced protein degradation in these cells (Koeprunner et al. 2001, DOI: 10.1101/gad.212401). One of the contributing factors for this phenomenon is for example presented in Kedde et al. 2007, DOI: 10.1016/j.cell.2007.11.034 , where we showed that miRNAs function is inhibited in the germ cells.

Reviewer #3 (Remarks to the Author):

The manuscript “Optochemical control over mRNA translation by photocaged phosphorodiamidate morpholino oligonucleotides in vivo” describes the use of photocaged guanidinium-linked morpholino (GMO)- phosphorodiamidate morpholino oligonucleotide (PMO) chimaeras to control protein translation. The core concept is based on the injection of two probes, which reduce synthetic economy as well as increase both complexity and potential side effects. Under irradiation, one is uncaged, displacing the one that blocks mRNA initiation translation site, thereby triggering protein translation. While the manuscript is comprehensive and includes exhaustive interesting biological data, particularly demonstrating the spatiotemporal control of translation, there are several concerns that limit its impact, especially for a high-profile journal like Nature Communications.

These are indeed important issues that we did not explain well enough, so we refer to them in detail at the beginning of the response letter and below.

The points we consider new in this paper are-

The principle of strand displacement for activation of translation *in vivo*, the design and synthesis of a new product that functions *in vivo*, the fact that one can use the same reagents for controlling the function of different RNAs, and the demonstration of efficient use in the complex and sensitive system of a developing embryo without induction of side effects.

We consider the procedure we report on as economical, since the two morpholino oligonucleotides are universal, allowing for large-scale production and, thus lowering costs. One of the morpholinos is a "standard control morpholino" available from Gene Tools (referred to in the text as tbMO). In contrast, systems tailored for individual PMOs require the generation of gene-specific compounds, which would significantly increase costs and require control experiments to test and exclude side effects. The currently published photocaged PMO synthesis procedure developed by one of the corresponding authors of this work (<https://www.nature.com/articles/nchembio.2007.30>), is used for inhibiting RNA translation rather than activating it and is very challenging, presumably a bottleneck for using it. Indeed, it is not possible to purchase a photomorpholino from “GeneTools”, which was the motivation for initiating the project.

The procedure for PMO synthesis we describe is adapted to the automated DNA synthesizer, making the synthesis easy. In the same direction, new chemistry has been developed for the synthesis of GMO by using very common and easily available cheap reagents like iodine and N-ethyl morpholine. The chemistry is developed in such a way that the synthesis of GMO was compatible with the synthesis of PMO so GMO-PMO was made by automated synthesizer in the present manuscript. PMOs are also nuclease resistant and thus easy to handle during purification. In addition, GMO-PMO does not require any additional transfecting agent like Endoportor and Lipofectamine used for PMOs and negatively charged oligonucleotides, respectively. These reduce the complexity and cost of the experiment and in the case of this study allow better distribution in the tissue.

1. First, the term "in vivo" may be somewhat misleading as the targeting appears to

be “ex vivo” and dependent on in vitro blockage that must be injected. This raises questions about the broader applicability of the technique (“universal technique”??), as the authors claim.

The solution we introduce into the embryo contains the RNA to which the translation-blocking morpholino is probably bound. The caged PMO is included in the mix as well. The term “*in vivo*” refers to the specific time and context in which the uncaging is induced, the translation-blocking morpholino is displaced, and protein expression begins. As presented, the procedure is “universal” in the sense of using it in organisms into which one injects RNA (e.g. *Drosophila*/*Zebrafish*/*Xenopus*). Adapting the idea to cell culture for example, would involve transfection of the reagents, while the photocaged PMO could be provided in the solution. As pointed out by the referee, the method we present would probably need adaptation for use in other systems. We thus exchange the word “Universally” to “Broadly”, which we hope delivers the message better.

2. a) Second, although the study introduces a never-reported approach, I struggle to see any significant advantage of using this methodology. On the contrary, if we compared this strategy with more straightforward techniques such as inducible promoters or photocleavable/photocaged siRNA, miRNA, PMOs

This issue was not clearly articulated in the submitted paper. It is an important point, and we address it at the beginning of the response letter before discussing other more specific concerns. We hope our explanation is clear, and we are happy to expand on it further if required.

b) In particular, the use of GMO-PMO conjugates adds unnecessary synthetic complexity compared to traditional photoprotected PMOs or other oligo-probes.

Based on our *in vivo* tests, we find that the GMO-PMO is twice as effective as the traditional PMO. We observed that compared to cPMO1 (traditional PMO), the photoactivation and induction of GFP expression were notably more effective when cPMO2 (GMO-PMO) was employed. We now specify in the text that, in the case of cPMO2, 89% of the injected embryos in which local expression was induced exhibited GFP expression. In contrast, only 45% of embryos treated with cPMO1 displayed GFP signal in the irradiated area.

c) The authors claim benefits from the GMO modification, but no compelling evidences are provided to support this. For example, while it is suggested that the GMO increases uptake, the probes are injected, and no data is shown to demonstrate improved cell permeability.

Along these lines, the authors need to demonstrate that the GMO modification increases (i) the affinity towards unmodified complementary PMO (as their data suggests a 2°C reduction compared to PMO alone, which indicates indeed reduced affinity), (ii) benefits the on/off kinetics of strand displacement and (iii) significantly improve cell-uptake in a human cell line model for example HeLapLuc705.

The ability of the GMO-PMO to be taken into the cell from the medium and be transferred from cell to cell is now presented in Figures S8 and S9.

In Figure S9, we transplanted cells that contain fluorescent Dextran (that marks the cells) and the different labeled PMOs into embryos that express mCherry on their membrane. What can be observed in Figure S9 is that the labeled PMO2 can get out of the transplanted

cells into neighboring cells that do not show the Dextran signal. Thus, while this phenomenon is observed in all cases, PMO2-BODIPY-488 can be transferred to adjacent, non-injected cells most efficiently (panel b).

Similar results can be observed in cells in culture (**point iii**). To examine the effect of the GMO modifications on the cellular permeability of the PMO2, we performed flow cytometry studies using the HeLa cell line (Figure S8 in the revised manuscript). Cells were exposed to various concentrations of BODIPY-tagged PMO1 and PMO2 to compare cellular uptake efficiency. After 4 hours of incubation, PMO2 exhibited 2-fold higher cellular permeability compared to PMO1.

Regarding point (I and ii)- Indeed, the decrease in T_m by 2°C degrees (to 34 degrees, well above the temperature at which zebrafish, *Xenopus*, and *Drosophila* experiments are performed) could be considered a disadvantage. However, In the context of the biological experiments, at this magnitude, this factor is not significant as the more important features are cell penetration and diffusibility within the cell. Thus, enhanced cell-penetrating capabilities and increased cytosolic availability of PMO2 compared to PMO1 are the basis for its increased biological activity.

The same holds for strand displacement kinetics parameters since the key issues relate to availability within the cell. The GMO modification facilitates the uptake of the compound into the cell by endocytic pathways and is transferred among cells by transcytosis (see Figure S9).

Relevant to this issue, in the case of PMOs (without the GMO modification), a synthetic agent such as Endoportor must be used for assisted uptake, which can interfere with cellular processes, mostly in an unfavorable way due to the associated toxicity of its constituent peptides.

3. Furthermore, the reduced affinity between PMO and RNA, compared to PMO/PMO-GMO hybrids, forces an excess of PMO, which diminishes the applicability of this method. Even with this excess of PMO/GMO, a significant portion of RNA/PMO duplexes remains intact, which limits the system's efficiency.

The design of the morpholinos and reporter RNAs is indeed such that the translation-blocking morpholino (tbMO) binds to the target RNA with a lower affinity than it does to the uncaged GMO-PMO. As explained in the methods- we lowered the affinity on purpose by mutating the morpholino-binding site in the reporter RNA, thereby reducing, rather than abolishing, the binding of the tbMO. The efficient inhibition is observed for example in Figure 3a.

As presented in Table 2, the doses used for the translational block as well as the concentrations of the cPMO are similar to those used for other translational blocking experiments in embryos/cells (Tarbashevich et al., 2015, <https://doi.org/10.1016/j.cub.2015.02.071>, Pradhan et al, 2021, 10.1038/s41467-021-26234-7, Ranjan et al., 2024, <https://doi.org/10.1038/s44319-024-00272-w>).

The homology between the tbMO and the uncaged GMO-PMO is perfect. Photo-uncaging of the cPMO/GMO-PMO leads to the restoration of the reporter signal by removing the translational block (Figure 3b). The gel shift assay shows that the PMO/RNA duplex remains even with an excess GMO-PMO, but these binding experiments report on the equilibrium state *in vitro*. In biological systems, the 'freed' mRNA is recruited to the translation machinery, which reduces its availability for binding the tbMO. We consider the system's efficiency high based on the biological results, with the gel shift assay serving as biophysical support.

In summary, while the data presented may be interesting and potentially publishable, the overall impact of this study does not meet the high standards expected for a journal like Nature Communications. I recommend redirecting this manuscript to a different journal more suited to the impact of this contribution, and suggest the authors provide a clearer justification for the use of such complex conjugates, supported by objective demonstrations of their benefits compared to simpler, existing technologies. The precedents in the field should also be included in the new manuscript version.

We hope the clarifications, explanations, and comparisons with other currently available options for inducing protein translation from introduced RNA convince the Referee of the power of the method. Concerning the general interest in methods for induction of RNA translation by light, we can only refer to the fact that currently such methods are not used in live embryos, and cell culture experiments where GFP expression is induced by light are published in key journals (e.g. Kloecker et al 2022, Nature Chemistry, <https://www.nature.com/articles/s41557-022-00972-7>).

Response to referees

We thank referees and editors for their work on our manuscript. Below we address individual comments of referees for the final revision round.

Reviewer #1 (Remarks to the Author):

The authors have done an excellent job addressing the key comments from my reviews. It's ready to be shared so that the true value (good and limits) of the technology will be identified based on different labs using this in diverse application areas.

We thank the referee for the positive feedback on our work.

Reviewer #2 (Remarks to the Author):

The revised manuscript is significantly improved. The authors have clarified the questions I have posed and made terminology within the manuscript consistent and easier for readers to follow. Furthermore, addition of quantitative measures of effects is important. I believe that this novel technique adds a new tool to study developmental biology questions, allowing controlled regional induction of protein expression.

We thank the referee for the positive evaluation of our work.

Reviewer #3 (Remarks to the Author):

The authors have addressed the concerns. Therefore, if the editor finds the impact of this work suitable for Nature Communications, I have no objections. However, before publication, please correct the statement regarding the precedent <https://doi.org/10.1038/s41557-022-00972-7>, as it has also been used in zebrafish <https://doi.org/10.1038/s42004-025-01411-7>.

We thank the referee for reviewing our manuscript.

We now cite the paper published last month in the introduction. We added a sentence to explain the related findings-

“ Upon irradiation, the translation machinery interacts with the RNA, and translation of marker proteins such as GFP and Luciferase can be induced in cells in culture and in zebrafish embryos (6,7,8). A biological effect was obtained when employing this approach under in vitro conditions, where induced H-Ras expression resulted in neurite expansion (6). Thus far, no reports have shown that this system can function in affecting biological pathways and developmental processes in whole animals. “